# Short-Term Skin Temperature Responses to Endurance Exercise: A Systematic Review of Methods and Future Challenges in the Use of Infrared Thermography

**DOI:** 10.3390/life11121286

**Published:** 2021-11-24

**Authors:** Daniel Rojas-Valverde, Pablo Tomás-Carús, Rafael Timón, Nuno Batalha, Braulio Sánchez-Ureña, Randall Gutiérrez-Vargas, Guillermo Olcina

**Affiliations:** 1Centro de Investigación y Diagnóstico en Salud y Deporte (CIDISAD), Escuela Ciencias del Movimiento Humano y Calidad de Vida (CIEMHCAVI), Universidad Nacional de Costa Rica, Heredia 86-3000, Costa Rica; randall.gutierrez.vargas@una.ac.cr; 2Clínica de Lesiones Deportivas (Rehab & Readapt), Escuela Ciencias del Movimiento Humano y Calidad de Vida (CIEMHCAVI), Universidad Nacional de Costa Rica, Heredia 86-3000, Costa Rica; 3Comprehensive Health Research Center (CHRC), Departamento de Desporto e Saúde, Escola de Ciências e Tecnologia—Universidade de Évora, 7000-727 Évora, Portugal; ptc@uevora.pt (P.T.-C.); nmpba@uevora.pt (N.B.); 4Grupo en Avances en el Entrenamiento Deportivo y Acondicionamiento Físico (GAEDAF), Facultad Ciencias del Deporte, Universidad de Extremadura, 10005 Cáceres, Spain; rtimon@unex.es (R.T.); golcina@unex.es (G.O.); 5Programa de Ciencias del Ejercicio y la Salud (PROCESA), Escuela Ciencias del Movimiento Humano y Calidad de Vida (CIEMHCAVI), Universidad Nacional de Costa Rica, Heredia 86-3000, Costa Rica; bsanchez@una.cr

**Keywords:** cycling, heat stress, marathon, running, thermal imaging

## Abstract

Background: Body temperature is often assessed in the core and the skin. Infrared thermography has been used to measure skin temperature (Tsk) in sport research and clinical practice. This study aimed to explore the information reported to date on the use of infrared thermography to detect short-term Tsk responses to endurance exercise and to identify the methodological considerations and knowledge gaps, and propose future directions. Method: A web search (PubMed, Science Direct, Google Scholar, and Web of Science) was conducted following systematic review guidelines, and 45 out of 2921 studies met the inclusion criteria (endurance sports, since 2000, English, full text available). Results: A total of 45 publications were extracted, in which most of the sample were runners (n = 457, 57.9%). Several differences between IRT imaging protocols and ROI selection could lead to potential heterogeneity of interpretations. These particularities in the methodology of the studies extracted are widely discussed in this systematic review. Conclusions: More analyses should be made considering different sports, exercise stimuli and intensities, especially using follow-up designs. Study-derived data could clarify the underlying thermo physiological processes and assess whether Tsk could be used a reliable proxy to describe live thermal regulation in endurance athletes and reduce their risk of exertional heat illness/stroke. Also more in-depth analyses may elucidate the Tsk interactions with other tissues during exercise-related responses, such as inflammation, damage, or pain.

## 1. Introduction

Current sport dynamics have evolved rapidly from a physical, physiological, technical, and tactical point of view [1]. With this growth, new technologies have emerged to control and monitor the different sports capacities related to optimal performance. Among these technologies, those that stand out due to their portability, easy access to information, innocuousness, relatively simple interpretation of data, and mainly due to their non-invasive nature have aroused the interest of stakeholders, such as athletes, medical staff, sports scientists, and physical therapists [2].

Examples of the technologies that have emerged as an option to control and monitor different phases of sports programming are tensiomyography [3], inertial measurement devices [4], locomotion tracking systems [5], mobile applications [6], oximeters [7], and heart rate monitors [8]. These technologies provide efficient and effective information for daily monitoring of the effect of changes in physical load prescription, the effectiveness of certain rehabilitation protocols, or the efficiency of recovery and sports physical readaptation processes.

Infrared thermography (IRT) is a valid tool that provides data on the heat radiated by a body by recording infrared emission within a spectrum of light invisible to humans [9]. In medicine, the applications of this technology have focused on the evaluation of physiological responses associated with skin temperature (Tsk), collaborating in the identification of a series of factors linked to cardiovascular, neurological, and oncological pathologies, and even more recently to areas of health and sports performance [10,11].

Among the monitored sports, high-volume and high-intensity sports have stood out, characterised by the great physical, physiological, and psychological stress they cause, which can trigger serious health problems for the athlete [12,13,14]. Of particular interest is the effect of heat on performance, but especially on the health of long-distance athletes, who commonly practice and compete outdoors [15]. Endurance sporting events are increasingly experiencing the consequences of climate change, specifically in the increase in temperature and relative humidity [16].

These adverse environmental conditions have an impact on the health of athletes related to thermal illnesses. Endurance sports practitioners may experience exertional hyperthermia and heat stress (e.g., rise in core temperature above 39 °C) when training or competing in warm-to-hot environments [17]. The environmental conditions described describe a potentially harmful scenario in endurance athletes, considering massive and elite events are being organised in hot and humid environments [18,19].

Some sports have attracted the attention of scientists worldwide, such as endurance disciplines, among other reasons for the massive growth in participation [1]. In this sense, a high component of volume load is required mainly due to a series of repeated eccentric actions that trigger neuromuscular and mechanical injuries in response to structural damage [8,20] and some heat issues due to rising global temperature [17]. Considering Tsk plays the fundamental role of regulating the heat exchange by convection, radiation, and evaporation [21], these alterations in Tsk could be due to metabolic responses, such as inflammation, potentially identified with infrared thermography [22,23]. Considering the above, it is necessary to explore the current evidence regarding the use of thermography for the health protection and injury prevention of endurance athletes. Therefore, the purpose of this systematic review was to explore the information reported to date on the use of IRT to detect short-term Tsk responses to endurance exercise, in order to propose future study directions on the application of thermography and identify those gaps in knowledge that need to be filled. 

## 2. Materials and Methods

A systematic review was performed following the Preferred Reporting Guidelines for Systematic Reviews and Meta-analyses (PRISMA) [24,25]. After conducting the search for studies, they were classified by year, identifying those that met the inclusion criteria for final consideration and extraction (see Figure 1). Two authors independently reviewed the manuscripts based on risk-of-bias. This assessment was made using a 4-point scale ranging from a low to high risk-of-bias qualification, and discrepancies between authors were resolved through consensus. Internal quality of each study was assessed using the Office of Health Assessment and Translation (OHAT) Risk of Bias Rating Tool [26]. The systematic review was conducted considering previously established guidelines, taking into account the delimitation of the research question, identification of relevant evidence, evaluation of the quality of the studies, summary of the results, and their interpretation [27].

### 2.1. Data Sources

An electronic literature search was performed in four different databases: PubMed (MEDLINE) (n = 221), Science Direct (EMBASE) (n = 948), Web of Science (WoS) (n = 319), and Google Scholar (n = 1433). The search was performed until 2:00 p.m. on 17 November 2021. The authors did not differentiate or discriminate by journal or manuscript authors. The search strategy considered Boolean phrases as search descriptors as follows: “thermography OR thermology OR thermometry OR thermal imaging OR skin temperature OR body temperature AND exercise OR endurance OR sports”. All references were extracted and imported into an open-source research tool (5.0.64, Zotero, Fairfax, VA, USA) to systematise studies.

### 2.2. Data Selection 

The following inclusion criteria were considered: studies containing keywords in the title or abstract, experimental designs in humans, studies published from 2000 to 2021 (November) in peer-review academic journals, studies exploring the effects on Tsk of practicing endurance sports, but not limited to swimming, cycling, running, skiing, triathlon, or kayaking. Studies written in the English language were considered. Other languages were also considered if a translation could be performed (e.g., Spanish, Italian, Portuguese). A single author collected the original database of studies and compiled them into a data spreadsheet (Microsoft Excel, Microsoft, Redmond, WA, USA). After duplicate removal, two independent authors analysed studies, considering inclusion and exclusion criteria. 

### 2.3. Data Collection and Extraction

Two different authors performed the analysis and selection of studies following the PRISMA protocol (see Figure 1). Specific exclusion criteria were followed to remove low-quality studies or those irrelevant to the primary aim of this systematic review, such as duplicates, articles with critical language limitations, studies in animals (e.g., dogs, horses), studies where the full text was not available, studies involving factors other than endurance sports, such as strength or conditioning research, different evaluation methods or technologies (e.g., thermistors, thermocouples, contact patches), book chapters, abstracts, conference papers and other technical reports, and articles which showed a severe lack of key information (e.g., participants information, discipline execute, confusing exercise protocol).

The protocol followed for selecting the studies was: a. an identification of potential studies, b. duplicate removal, c. title, abstract, and year examination, d. method quality and relevance review, and e. exploration of full texts. Finally, f. those studies with a lack of information (e.g., lack of sample data, lack of technical characteristics of equipment) were excluded. 

The finally selected studies were analysed, and the following data were extracted using a descriptive table (see Table 1): authors and year of publication, study design and task performed, results and main study outcomes. Besides, Table 2 and Table 3 present methodological relevant data of each study considering the Thermographic Imaging in Sports and Exercise Medicine checklist (TISEM) [28] as follows: participants and environment characteristics (participants data, previous instructions, extrinsic factors, environmental condition, environmental setup, equipment, image background, thermal adaptation) (see Table 2); and camera settings and setup (camera preparation, image recording, camera position, emissivity, assessment time, body position, method of drying the skin and image evaluation) (see Table 3).

## 3. Results

Table 1, Table 2 and Table 3. shows the compilation of the analysis of the 45 selected studies for this systematic review; they describe the study’s design, analysis, and main outcomes (Table 1), methodological aspects based on TISEM related to participant and environment characteristics (Table 2), and camera settings and setup (Table 3).

### 3.1. Study’s Protocols, Analysis, and Main Outcomes

In respect to study design, most of the studies explored acute effects of exercise (n = 41, 92.2%) and just a few analysed Tsk using chronic-based designs (n = 4, 8.8%). Besides, the most studied sport was running (n = 26, 57.7%), followed by cycling (n = 14, 28%), and others, such as swimming and kayaking (n = 5, 10%). A total of 42 protocols used a cross-sectional design, and the studies measured pre–post temperature changes during acute stimulus, such as running a marathon (n = 3, 6.7%), half-marathon (n = 1, 2.2%), 30 km (n = 1, 2.2%), an incremental test until exhaustion (n = 2, 4.4%), and other treadmill tests during 12 min (n = 1, 2.2%), 15 min (n = 3, 6.7%), 30 min (n = 4, 8.8%), 35 min (n = 1, 2.2%), 40 min (n = 1, 2.2%), and 60 min (n = 6, 13.3%). Other studies used cycling until exhaustion on an ergometer (n= 5, 11.1%), cycling during 6 min (n = 1, 2.2%), 45 min (n = 1, 2.2%), 60 min (n = 1, 2.2), or 90 min (n = 2, 4.4%), and swimming (n = 2, 4.4%) and kayaking (n = 1, 2.2%) for 1000 m. All stimuli were performed with a great variety of intensities.

Most of the papers performed a bilateral ROI analysis (n = 34, 75.6%), but unilateral exploration was also performed (n = 11, 24.4%). Moreover, the ROIs were analysed considering time-points alone (n = 17, 37.8%), or taking into account some condition (n = 14, 31.1%), laterality (n = 9, 20%), or differences between ROIs (n = 2, 4.4%), and also sex (n = 3, 6.7%). Moreover, a few studies explored the correlation between Tsk and other mechanical and physiological parameters (n = 2, 4.4%). Two studies (n = 2, 4.4%) selected the ROIs using an objective statistical technique, such as the principal component analysis (see Table 1). 

The authors selected between 1 and 28 different ROIs from anterior and posterior upper and lower limbs, trunk, chest, back, forehead, and others. From the total of the 45 studies, the predominant analysis was based on lower limbs ROIs (n = 19, 42.2%) and other regions, such as upper and lower limbs (n = 14, 31.1%), feet and soles (n = 8, 17.8%), upper limbs (n = 6, 13.3%), face, or inner canthus (n = 4, 8.9%) (see Table 1).

Finally, the main results of the studies showed that, during exercise, the Tsk decreases (n = 12, 26.7%), and after the effort, during recovery, increases (n = 19, 42.2%); there were no lateral asymmetries due to exercise (n = 5, 11.1%); there was an inverse relationship between core and Tsk during exercise (n = 3, 6.7%); Tsk could be influenced by body composition, muscle activity, and sweating rate (n = 5, 11.1%); there is no consensus in the differences by sex (n = 3, 6.7%); and compression gear had no influence on Tsk (n = 1, 2.2%) (see Table 1).

### 3.2. Participant and Environmental Characteristics

A total of 45 studies were extracted analysing the data of 788 (656 male and 132 female) participants. The majority of the sample were runners (n = 457, 57.9%), followed by cyclists (n = 144, 18.2%), swimmers (n = 48, 6.1%), kayakers (n = 19, 2.4%), sailors (n = 18, 2.3%), triathletes (n= 10, 13%), and skiers (n = 26, 3.3%); other participants were only described as active (n = 66, 8.4).

A total of 22 (48.9%) out of 45 studies reported following previous instructions, such as no diuretics, drug, tobacco, alcohol, caffeine, heavy meals, or tea intake before the thermograms were taken. Moreover, no sunbathing, UV ray exposure, exercise, creams, or lotions were allowed. An additional six studies (0.7%) reported as exclusion criteria present skin burns, kidney problems, symptoms of pain, osteomioarticular injury, using compression garments, or attending physical therapy or using recovery methods prior to the evaluations.

With respect to the thermographic camera, 18 different models of different brands were used: FLIR (n = 37, 82.2%), Fluke (n = 3, 6.7%), AVIO (n = 2, 4.4%), DALI (n = 1, 2.2%), PCE (n = 1, 2.4%), CEDIP (n = 1, 2.2%), and NEC (n = 1, 2.2%). The preferred devices were the FLIR E60 (n= 11, 24.4%) and FLIR E60bx (n = 6, 13.3%), FLIR T335 (n = 3, 6.7%), FLIR T420 (n = 3, 6.7%), and models. 

The authors used a wide range of camera and environmental settings to acquire the thermograms. Thermal adaptation was performed during 10 min (n = 22, 48.9%), 15 min (n = 7, 15.6%), 20 min (n = 1, 2.2%), and 30 min (n = 2, 4.4%); another eight (17.8%) articles did not report this data. 

Moreover, some room environmental conditions were reported as temperature and humidity in 84.4% of the cases (n = 38), ranging from 18.0 to 30.0 °C and 27.9 to 80% of humidity. Besides, a total of 20 (44.4%) studies reported avoiding direct airflow, electronic equipment (e.g., heating/cooling air conditioning systems) near the evaluation area, and artificial or natural light. Finally, 16 (35.6%) studies reported using an antireflective panel as a background of the thermograms to avoid temperature bias.

### 3.3. Camera Settings and Setup

Regarding the camera preparation, such as calibration (e.g., a reference plate connected to a thermistor), turned-up time, and stabilisation, only 20 (44.4 %) studies reported the procedure followed. There was a lack of information regarding the specifics of these procedures, which does not allow replication.

The thermographic camera was set 1 m (n = 10, 22.2%), 1.5 m (n = 4, 8.9%), 2 m (n = 1, 2.2%), 2.5 m (n = 1, 2.2%), 3 m (n = 4, 8.9%), 3.5 m (n = 1, 2.2%), 4 m (n = 1, 2.2%), and 6 m (n = 1, 2.2%) from the skin or region of interest. Another 20 (48.8%) protocols did not report the camera distance from the participant.

The participant’s position during the IRT protocol was not reported in only two of the protocols. In addition, a total of 40 (88.9%) of the protocols used the average of ROIs Tsk to analyse the data.

## 4. Discussion

This study aimed to explore the information reported to date on using IRT to detect short-term Tsk responses to endurance exercise to propose future study directions in the application of thermography and identify those gaps in knowledge to be filled. A total of 41 publications, including a sample of 660 participants, were extracted. 

### 4.1. Methodological Considerations to Assess Tsk

The most studied sport was running, followed by cycling. Studies analysed Tsk using cross-sectional designs through acute pre–post exercise protocols lasting from 15 to 90 min of running, cycling, swimming, and kayaking. The thermograms were collected using 19 different cameras under different environmental conditions and settings. Most of the authors preferred to set the IRT camera 1 m from participants’ skin, using a 10 min thermal adaptation with a temperature of 18.0–27.0 °C and 50.0–64.0% relative humidity. Moreover, the ROIs selected for analysis varied widely, and only one study used a statistical technique (e.g., principal component analysis) to identify the most relevant ROIs.

Although there are some guidelines regarding the optimal settings that should be followed to obtain better thermogram results [9,10,11], this systematic review has shown that some studies did not report crucial data, such as room temperature, humidity, or thermal adaptation time. Considering multiple thermographic brands and settings selected to obtain the thermograms, studies should be performed on how different settings, such as room temperature and relative humidity, the camera distance from the skin, thermal adaptation time, and participants’ positions, could potentially influence temperature data. How do the stimuli influence temperature changes depending on the exercise settings (e.g., exercising in hot, humid, windy environments) [31,58]?

Additionally, considering the studies extracted, the selection of several ROIs sometimes did not consider the segmentation of an area (e.g., quadriceps into rectus femoris, vastus lateralis, and medialis); it should be analysed whether the sub-areas of a body segment could present differences in temperature due to the different ROIs selected [35]. 

There is a lack of objectivity when selecting the ROIs to be analysed. This is one of the main concerns; usually they are based on technical considerations, such as the role of the area in the sport mechanics and performance. Some statistical data mining techniques that have been used in other sport and exercise science studies to discriminate the most relevant variables to be analysed could be used to select the ROIs [35]. One of the preferred statistical methods is principal component analysis. Still, other options (e.g., Bat algorithm, MLP networks, Bayesian classificatory, SMO algorithm, random forest, logistic regression, Levenberg–Marquart algorithm, K-means grouping algorithm, hierarchic grouping) are also valid and applicable to this kind of data.

Moreover, despite the vast amount of evidence regarding the Tsk responses in acute conditions to relatively short stimuli, such as time to exhaustion tests, incremental tests, and other similar tests lasting 15–60 min at different intensities, there is a lack of studies recording the physiological skin responses to heavier and longer stimuli, such as the marathon, long-distance triathlons, multi-stage cycling, open-water swimming, and other similar efforts, that could allow a more in-depth analysis of the behaviour of Tsk and its relationship with other well-known physiological responses, such as muscle damage, pain, and temperature regulation [31,58].

The preferred acute stimuli used to explore Tsk change using thermography were a 60 min aerobic test in running (n = 7) and an incremental test until exhaustion in cycling (n = 5). To better understand the Tsk responses after exercising, more longitudinal studies are needed to elucidate the physiological cascade influencing Tsk and how it is related to adjacent tissue responses [73]. This may allow scientists, researchers, athletes, medical staff, coaches, and other stakeholders to detect, monitor, and control the Tsk changes in endurance practitioners at both the group and the individual level. This is key for extending the knowledge of the physiological effects of endurance exercise beyond a single moment in time.

Some methodological guidelines are available as TISEM [28] to report critical methodological procedures when using IRT in sports and exercise sciences. Indeed, there is a lack of information in the studies selected considering the participants and environment characteristics, and camera settings and setup. Future studies must follow this kind of methodological guidelines to make the studies replicable.

### 4.2. Challenges in the Use of IRT in Endurance Sports

Researchers have been concerned with understanding how exposure to prolonged and strenuous exercise could impact short- and long-term performance and health, physiologically and physically [12]. Moreover, there has been a great increase in endurance running participation [74], mostly among increasingly younger participants commonly requiring medical assistance while participating in these endurance events [75]. Following this information, thermography has been shown to detect acute changes in Tsk provoked by exercise. During physical movement, the skin blood vessels contract slightly due to the higher muscle activity as a response to higher sympathetic activity [76], which is why this systematic review identified articles suggesting a decrease in Tsk during effort. In contrast, after exercise, an increase in Tsk is expected due to the heat produced by muscle contraction, when the cutaneous vessels may dilate and dissipate heat. In this regard, more studies are needed to understand how the control circuit of heat conduction and dissipation from the muscles to the skin works and how these physiological processes could later impact Tsk [77]. Moreover, more evidence is needed regarding the interaction between core and peripheric tissue temperature with Tsk, since there are several factors influencing Tsk, such as blood flow rate, thermal conductance, thermal capacitance, skin surface area, body segment lengths, and metabolic rates [78].

Skin is the outer layer that controls thermoregulatory functions and heat transfer for sweating, vasodilation and vasoconstrictions, and shivering. Although, a better understanding of the functionality of perforator and perforasome (potential thermal connection to other nodes by convection and conduction) as the interconnected vascular system between the skin and deeper tissues during pain, inflammatory cascade, and tissue damage as regular outcomes after prolonged exercises is needed [67]. Consequently, multi-segment and multi-node (e.g., core, muscle, fat, skin) thermal modelling studies could be performed for endurance exercising [78], considering Tsk change during endurance sports could oscillate [76] while intramuscular temperature remains constant [79]. Studies that clarify these dynamics and physiological interaction between tissues are necessary, as well as the understanding of how Tsk may or may not suggest changes in structures at a deeper level.

In line with the abovementioned, in endurance sports, it seems that the rise in Tsk could be due to the heat generated by the prolonged exercise and its subsequent physiological processes as the increase in endothelial nitric oxide, glycogen resynthesis, or increase in systemic hormones [67]. Despite there being a need for a more in-depth analysis of the thermo physiological responses to endurance exercise, skin blood flow and Tsk play a critical role in thermoregulatory processes. Due to the prolonged duration of these, heterogeneous behaviour in Tsk that could be found in the different ROIs could be explained by the different environmental conditions, exercises durations, intensities, and mechanical load [76]. The permanent switch from skin vasoconstrictions to vasodilation and vice versa, due to the need to lose or maintain body temperature heat, could provoke a decrease and subsequent increase in Tsk. This could be the reason why there could be changes in Tsk during a constant-load exercising by time. Considering these continuous and heterogenous changes in Tsk, there is a need to explore new technology that allows monitoring of the athlete constantly. As has been evaluated so far, recording the temperature before and after an endurance event does not seem to be sufficient to record these potential changes throughout an event of long duration and relatively variable efforts (e.g., trail running, kayak, open-water swimming, cross-country skiing).

Tsk requires a standardised protocol commonly considering environmental conditions, calibration, and consistent set-up for the thermographic camera [80]. All these protocols were selected for laboratory-based research, and no field protocol have been proposed due to some set-up limitations. In this sense, considering that endurance sports are usually practiced outdoors, IRT have a great limitation to outpatient and field-based assessments. In this sense, the TISEM [28] methodology guidelines may be followed to standardise the reporting of key imaging protocols, equipment characterisation, participant preparation, and temperature assessment techniques. A revised protocol should be proposed in future studies to evaluate Tsk using IRT in field-based settings.

### 4.3. Limitations

While this study aimed to explore the knowledge reported to date on the use of IRT to detect short-term Tsk responses to endurance exercise, the results and outcomes must be seen in the light of some limitations. One of the main limitations is the wide range and difference between study designs, assessment settings, and temperature. This issue makes the studies and outcomes difficult to compare and interpret, and, finally, difficult to link with other evidence and combine to obtain more precise results. Considering the organic characteristics of the sample and that endurance sports participants are relatively uncommon, having a sufficient sample size or follow-up designs is challenging. More studies need to be performed regarding the sample characteristics considering a more diverse population in order to have more generalisable results to then propose a more global solution. 

## 5. Conclusions

Although there is the relatively large number of published articles regarding the use of IRT to assess acute Tsk responses to endurance exercise and sports practice, several differences between IRT imaging protocols and ROI selection could lead to potential heterogeneity of interpretations. More in-depth analysis should be made considering different sports, exercise stimuli, and intensities, especially using follow-up designs and multi-segment and multi-node models.

An objective method is needed to select the most relevant ROIs when analysing sport-related responses to exercise considering the muscle activity and joints involved in a specific movement. Hand in hand with standardisation of Tsk assessing protocols using IRT, more in-depth analysis-derived data could clarify the underlying physiological processes involved in the regulation of Tsk and its interaction with other tissues (e.g., adipose tissue, muscle) during endurance-exercise-related responses, such as inflammation, damage, or pain, mainly the role of perforasome in Tsk regulation over time.

Finally, new technology in the monitoring of the Tsk and in the data analysis (e.g., software) is necessary to have accurate information on the physiological state of the athlete and, thus, be able to make decisions in a timely manner. 

## Figures and Tables

**Figure 1 life-11-01286-f001:**
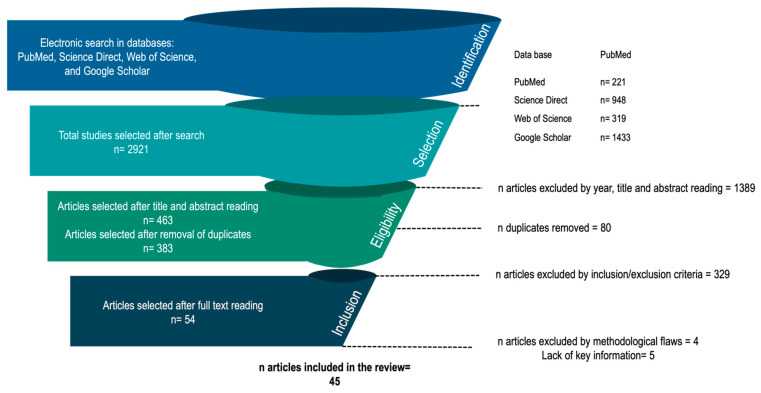
Flow diagram of the study identification, selection, and inclusion.

**Table 1 life-11-01286-t001:** Articles included in the systematic review considering fundamental study design, analysis, and main outcomes.

#	Author/Year of Publication	Task/Design	Body Region	ROIs Selection	Data Analysis	Outcomes
1	Tumilty et al., 2019 [29]	1 × 9 weeks Thermograms: once/day	Bilateral Achilles	Rectangle 10 × 40 pixels from the superior border of the calcaneus	Laterality * weeks	No bilateral or between weeks variations (0.50 ± 0.43 °C) in Achilles Tsk.
2	Gil-Calvo et al., 2020 [30]	2 × 15 min run (control vs. provoked asymmetry (1.5 kg ankle weight)) Thermogram: pre and immediately post run	Bilateral foot	6 ROIs feet (complete soles, forefoot, midfoot, rearfoot (100%, 50%, 19%, and 31% of foot sole length feet, respectively), hallux, and toes.	Condition * time-points * Laterality	No bilateral differences despite group. Asymmetrical running provoked higher Tsk.
3	Gutiérrez-Vargas et al., 2017 [31]	1 × marathon Thermogram: 3 days pre, immediately post and 24 h post	Bilateral lower limbs	14 ROIs (8 anterior and 6 posterior)	Time-points * Laterality	No bilateral difference. Time-points differences in almost all ROIs, >1 °C in the knee, vastus medialis, vastus lateralis, rectus femoris, adductor.
4	Fournet et al., 2013 [32]	1 × 40 min run 70% VO_2_max in 10 °C and 54.76% relative humidity Thermogram: pre (rest), post run (10 min), post run 2 (40 min), post	Bilateral anterior and posterior, upper and lower body	11 morphed ROIs	Time-points * sex	Females lower Tsk than males, no skinfold thickness influence.
5	Priego-Quesada et al., 2016 [33]	2 × 45 min cycling (35% and 50% of peak power output and cadence 95 rpm). Thermograms: Pre, immediately post, and 10 min post	Unilateral anterior and posterior	17 ROIs (deltoid, chest, abdomen, upper back, lower back, vastus lateralis, rectus femoris, abductor, vastus medialis, biceps femoris, semitendinosus, knee, popliteal, tibialis anterior, gastrocnemius, ankle anterior, and Achilles)	Time-points * cycling workout * ROI	Increase in Tsk post cycling in knee extensors and decrease in trunk. After 10 min post posterior, Tsk of lower limb and trunk increased. Inverse relationships were observed between core and Tsks.
6	Priego-Quesada et al., 2015 [34]	2 × 30 min run at 75% of maximal aerobic speed (control and compression stockings) Thermograms: pre and immediately post and 10 min post	Unilateral lower limb	12 ROIs (tibialis anterior, ankle anterior, and gastrocnemius and vastus lateralis, abductor. and semitendinosus)	Time-points * condition * ROI	Compression stockings increase Tsk in the regions in contact and not in contact with the garment.
7	Priego-Quesada et al., 2016 [35]	45 min cycling at 50% of peak power at 90 rpm of cadence (control and fitted position) Thermogram: Pre, immediately post, and 10 min post	Bilateral anterior and posterior, upper and lower body	16 ROIs (chest, abdomen, upper back, lower back, vastus lateralis, rectus femoris, abductor, vastus medialis, biceps femoris, semitendinosus, knee, popliteus, tibialis anterior, gastrocnemius, anterior ankle, and Achilles.	Principal component analysis. Condition * time-points	Factor analysis is a useful method to determine a lower number of ROIs. Differences between groups of ROIs were related to tissue composition, muscular activity, and capacity for sweating.
8	Priego-Quesada et al., 2015 [36]	1 × cycling test to exhaustion. Thermograms: pre and immediately post, and 10 min post	Unilateral Lower limbs	4 ROIs, (gastrocnemius, rectus femoris, 2 × biceps femoris).	Correlation with EMG and time-points differences.	Tsk of knee extensors increases after cycling. Vastus lateralis overall activation was inversely related to Tsk.
9	Priego-Quesada et al., 2019 [37]	3 days of training cycling and swimming. Thermograms: pre, 2nd day, 3rd day	Bilateral upper and lower limbs	8 ROIs (arms, anterior and posterior lower limbs)	Time-points * laterality	Tsk increased after training for most of the body regions. Tsk variation was related to muscle mass and weekly training volume.
10	Hadžić, et al., 2019 [22]	1 × 6 min cycling (100 W) + stretching and hamstrings isokinetic exercise (exercising vs no exercising limb) Thermograms: Each 30 s of video	Bilateral	Quadriceps (vastus medialis)	Correlation Tsk and power. Condition difference.	Negative correlation between Tsk change and muscle power output.
11	Fernandes et al., 2016 [38]	1 × 60 min of at 60% VO_2_max Thermograms: 12 measurements every 5 min	Unilateral	Inner canthus	Time-points differences	Poor agreement between core temperature and inner canthus temperature.
12	Rodriguez-Sanz et al., 2019 [39]	1 × Running 15 min on a treadmill at a speed of 8 km/h Thermograms: Pre and post	Bilateral	Gastrocnemius	Time-points differences	Runners with functional equinus condition presented a higher Tsk of gastrocnemius after a light running activity.
13	Priego-Quesada et al., 2020 [40]	1 × 30 min treadmill run (8 km/h increasing 1 km/h every 30 s) at 1% slope at 12/20 Borg Thermograms: pre and post	Bilateral	Anterior and posterior thighs	Time-points * laterality	Attaching one thermal contact sensor throughout the protocol and another only a while before each data acquisition is a good option for studying the effect of sweat accumulation on Tsk measurement.
14	Merla et al., 2010 [41]	1 × treadmill running until reaching individual maximum heart rate or voluntary interruption Thermograms: pre and post	Unilateral	6 ROIs: forearm, pectoral, mammary, sternal, abdominal, and thigh.	Time-points differences	The Tsk decreased during exercise and increased in recovery phase.
15	Luo et al., 2015 [42]	1 × 30 min running at 8 km/h Thermograms: pre and post	Unilateral	10 ROIs: Medial foot: medial forefoot, medial mid-foot, medial hind-foot, and medial ankle. Instep: fore instep, hind instep. Lateral foot: lateral forefoot, lateral mid-foot, lateral hind-foot, and lateral ankle	Time-points differences	Foot temperature increased exponentially until 15 min of exercise when the increasing rate slowed down.
16	Rynkiewicz et al., 2015 [43]	1 × 1000 m all-out paddling in kayak ergometer Thermograms: pre and post	Bilateral	3 ROIs: anterior trunk, shoulders; and posterior shoulders.	Time-points differences	Decrease in superficial temperature. Advanced kayakers presented greater differences.
17	Robles-Dorado 2016 [44]	1 × 30 km running Thermograms: pre and post	Bilateral	ROIs: foot sole, tibialis anterior, quadriceps, calf, and hamstrings.	Time-points differences	Increased temperature in foot sole, quadriceps, and Achilles, no variations in Tsk of semitendinosus, semimembranosus and tibialis anterior.
18	Sanz-López et al., 2016 [45]	2 d/week during 6 week eccentric training Thermograms: pre–post an hour running at 1 h at 80% of maximal heart rate	Bilateral	Achilles and patellar tendons	Group * time-points	Eccentric overload training causes particular adaptations in tendon tissues.
19	Priego-Quesada et al., 2016 [46]	3 (different saddle height) × 45 min cycling at individual 50% peak power output at 90 rpm of cadence	Unilateral	16 ROIs in trunk and lower limbs	Group * time-points	Different postures assumed by the cyclist due to different saddle height did not influence temperature measurements.
20	Ludwig et al., 2016 [47]	1 × incremental cycling test until exhaustion (100 W, 1 min increases of 25 W, 80–90 rpm of cadence).	Bilateral	Left and right thigh	Time-points differences	Tsk dynamic of quadriceps showed an explicit decrease during an incremental maximal exercise and a subsequent rapid recovery immediately after exhaustion.
21	Priego-Quesada et al., 2017 [48]	1 × Incremental cycling test to exhaustion (105 W, increases of 35 W each 3 min, 55 rpm of cadence). Thermograms: pre, after 10 min, post	Bilateral	4 ROIs: Vastus Lateralis, Rectus Femoris, Biceps Femoris, and Gastrocnemius Medialis	Group * time-points	Tsk was positively correlated with peak power output and heat production. At higher physical fitness, higher heat production and higher Tsk.
22	Priego-Quesada et al., 2017 [49]	1 × 30 min running (10 min at 60% of maximal aerobic speed and 20 min at 80%). Thermograms: pre–post	Bilateral	4 ROIs in foot sole.	Medio-lateral differences	Tsk is not related to foot eversion.
23	Jiménez-Pérez et al., 2020 [50]	2 × 30-min running at 75% of VO_2_max Thermogram: pre–post	Unilateral	10 ROIs: Plantar surface of dominant sole of the foot	Gender differences	Foot orthoses do not modify plantar surface temperature after running in healthy runners of either gender.
24	Mendonca-Barboza et al., 2020 [51]	1 × Cooper’s 12-min run test Thermograms: pre–post	Bilateral	4 ROIs: anterior and posterior views of the trunk and upper limbs, and anterior and posterior views of the lower limbs	Laterality * time-points	Tsk change of middle-distance runners was symmetrical between sides, decreasing in upper limbs and trunk and increasing in lower limbs after a short-term maximum effort test.
25	Duygu et al., 2019 [52]	1 × ergometer running test until exhaustion (11.3 km/h, increases of 2 ° every min) Thermograms: pre–post	Bilateral	Quadriceps and hamstrings	Group * time-points	Temperature change after anaerobic performance was not significant.
26	Pérez-Guarner 2019 [53]	1 × Half-Marathon competition at world championship Thermograms: pre (48 h), pre (24 h), post (24 h), and post (48 h).	Bilateral	ROIs upper and lower limbs	Time-points differences	Tsk responses to a half-marathon were not able to predict physiological stress markers.
27	Drzazga et al., 2018 [54]	1 × an hour running (individual lactate threshold intensity) Thermograms: pre–post	Bilateral	22 ROIs: upper body and lower limbs	Group * Time-points	Significant decrease in upper body temperature in skiers and increase in lower limb temperature in swimmers.
28	Trecroci et al., 2018 [55]	1 × maximal incremental cycling test (100 W, increases of 25 W/min until exhaustion, 90 rpm cadence) Thermograms: pre and immediately post	Bilateral	Thighs	Laterality * Time-points	Bilateral Tsk did not show any differences. No relation between asymmetry of Tsk with muscle effort.
29	Novotny et al., 2017 [56]	1 × 1000 m all-out crawl swimming Thermograms: pre–psot	Bilateral	20 ROIs: deltoids anterior, posterior and lateralis, rhomboids major and minor, pectoralis major and minor, erector spinae, latissimus, trapezius, triceps brachii, and biceps brachii	Laterality * time-points	Significant increase in triceps brachii, deltoids temperature.
30	Novotny et al., 2015 [57]	Breaststroke swimming 1000 m as fast as possible Thermograms:	Bilateral	20 ROIs: deltoids anterior, posterior and lateralis, rhomboids major and minor, pectoralis major and minor, erector spinae, latissimus, trapezius, triceps brachii, and biceps brachii	Laterality * time-points	Significant increase in Tsk of deltoideus and triceps. Right–left difference in temperatures was not significant.
31	Priego-Quesada et al., 2020 [58]	1 × marathon Thermograms: pre (48 h), pre (24 h), post (24 h), and post (48 h).	Bilateral	Lower limbs	Time-points differences	Baseline Tsk was not altered 24 or 48 h after a marathon.
32	Requena-Bueno et al., 2020 [59]	1 × 30 min running (80% maximum aerobic speed on a treadmill with a 1% slope) Thermograms: pre–post	Bilateral	9 ROIs: hallux, toes, medial metatarsal, central metatarsal, lateral metatarsal, medial midfoot, lateral midfoot, medial heel, and lateral heel	Time-points * laterality * analysis procedure	Analysis using ThermoHuman resulted in a reduction of 86% in the time required to process the thermograms.
33	Bertucci et al., 2013 [60]	1 × Incremental cycling test (4 min at 100 W, increases every 4 min by 40 W until exhaustion) Thermograms: pre–post	Bilateral	Lower limbs (thigh)	Time-points differences	Relation between increase in gross efficiency and Tsk.
34	Ferreira-Oliveira et al., 2018 [61]	1 × progressive cycling test (up to 85% of Hrmax, 50 to 60 rpm of cadence at 20 W, 15 W increases every 2 min until voluntary exhaustion) Thermograms: 15 min during and 60 min recovery (after)	Bilateral	ROIs: thighs, legs, arms, forearms, upper back, lower back, chest, and abdomen	Time-points differences	Decrease in temperature in chest, abdomen, upper back, lumbar region, anterior and posterior thigh, anterior and posterior leg. Temperature increased after 15 min recovery.
35	Andrade-Fernandes et al., 2016 [62]	1 × 1 h of treadmill running at 60% of the VO_2_max. Thermograms: every 5 min (12 times)		28 ROIs: forehead, face, chest, abdomen, back, lumbar, anterior and posterior neck, and posterior and anterior views of the right and left hands, forearms, upper arms, thighs, and legs	Time-points differences	Significant changes in Tsk due to running.
36	Akimov & Son’kin 2011 [63]	1 × stepwise ergometer test (60 W with increases of 60 W each 2 min, constant cadence: 60 rpm) (endurance and multisports) Thermograms: video every 30 s	Single area	Forehead	Conditions * group	Endurance and multisports group’s Tsk decreased until exhaustion.
37	Cholewka et al., 2016 [64]	1 × incremental test (50 W with increases of 30 W each 3 min) Thermograms: video every 180 s	Unilateral	5 ROIs (face, chest, arms, back, calf)	Time-points differences	Decrease in Tsk over time during exercise.
38	Tanda 2018 [65]	2 × 30 min treadmill runs (constant (6 km/h) vs. graded load (1.5 km/h increases every 5 min until 13.5 km/h was reached)) Thermograms: each 5 min	Bilateral	18 ROIs (upper and lower limbs, chest, back, face)	Time-points * condition	Variations over time in Tsk in both conditions. Tsk was reduced the first 10 min of exercise.
39	Crenna & Tanda 2020 [66]	1 × 60–90 min treadmill run at 10.2–14 km/h (90–95% of max) Thermograms: each 5 min	Bilateral	14 ROIs (chest, abdomen, lower limbs, upper limbs, back)	Time-points difference	Large heterogeneity depending on the ROI during exercise.
40	Rojas-Valverde et al., 2021 [67]	Prolonged running (marathon) Thermograms: pre (15 d and 45 min), post (24 h and 6 d)	Bilateral Anterior–posterior	13 ROIs (lower limbs)	Time-points differences Correlation with muscle damage markers	Tsk increased the day after the marathon and no relationships observed between muscle damage markers and Tsk.
41	Fernández-Cuevas et al., 2014 [68]	1 × 45 min treadmill run at 60–75% heart rate max Thermograms: pre, immediately post, and 60 min post	Bilateral upper and lower limbs	71 ROIs	Time-points differences	Tsk decreases and increases immediately post exercise depending of the ROI but, during recovery, Tsk usually increases.
42	Racinais et al., 2021 [69]	Marathon and race-walk (20–50 km) Thermograms: pre, immediately post	Bilateral upper and lower limbs	18 ROIs (neck, chest, shoulder arms and legs)	Time-points differences	Lower pre-race Tsk correlated with faster finished times. DNF athletes presented higher pre-race Tsk.
43	Machado et al., 2021 [70]	30 min run, 1% slope self-selected speed Thermograms: pre, immediately post	Bilateral	7 ROIs (sole and lower limbs)	Time-points * between devices difference	C2 and Flir-One pro presented lower mean and maximum Tsk than E60Bx. High data variability between cameras.
44	Binek et al., 2021 [71]	60 min running on treadmill with 80% of VO_2_max Thermograms: pre, imediately post, and 10 min recovery	Bilateral	4 ROIs (lower limbs)	Time-points * sex	Tsk of females is lower than males, Tsk changes due to exercise were greater in women.
45	Jones et al., 2021 [72]	Two middle distance runners Thermograms: 42 days observations	Bilateral	4 ROIs (lower limbs)	Time-points differences	No changes in daily Tsk.

**Table 2 life-11-01286-t002:** Articles’ methodological (participants and environment characteristics) considerations following the Thermographic Imaging in Sports and Exercise Medicine [28] checklist.

#	Reference	Participants Data	Previous Instructions	Extrinsic Factors	Environmental Conditions	Environmental Setup	Equipment	Image Background	Thermal Adaptation
1	Tumilty et al. [29]	10♂ and 7♀ (18–25 years), competitive cross-country runners (training: 25 miles/w)	No alcohol, caffeine, smoking, or exercise	NR	T°: 21.8 ± 0.5 °C %RH: 47.3 ± 8.1%	Minimized airflow	FLIR T450SC Accuracy: 1% Sensitivity: ≤0.05 °C	Non-reflective surfaces	15 min (seated)
2	Gil-Calvo et al. [30]	17♂ (27.0 ± 8.0 years), recreational runners (training: 4.0 ± 2.0 sessions/w)	No alcohol, caffeine, smoking, tea, drugs, heavy meals	NR	Assessed, no data	Reflected temperature measured. No electronic devices, light, or airflow in the room	FLIR E60bx Accuracy: 2% Sensitivity: <0.05 °C	Black panel	10 min (seated, extended knees)
3	Gutiérrez-Vargas et al. [31]	10♂ and 7♀ (35.8 ± 7.0 years) recreational runners (training: 9.3 ± 6.6 years of experience)	NR	NR	T°: 22.3 ± 0.9 °C %RH: 64.7 ± 8.3%	NR	FLIR T440 Accuracy: 2% Sensitivity: ≤0.2 °C	NR	10 min (biped)
4	Fournet et al. [32]	9♂ and 9♀ (18–25 years), physically active Caucasians	NR	NR	T°: 22 °C	NR	FLIR Thermacam B2 Accuracy: ±2% Sensitivity: ±0.1 °C	NR	10 min
5	Priego-Quesada et al. [33]	14♂ (29.9 ± 8.3 years), cyclists (training: 162 ± 77 km/w)	No alcohol, caffeine, smoking, sunbathing, UV ray exposure, heavy meals, or exercise	NR	Controlled not specified	No electronic equipment or persons	FLIR E-60 Accuracy: 2% Sensitivity: ≤0.05 °C	Anti-reflective panel	10–15 min
6	Priego-Quesada et al. [34]	10♂ and 14 ♀ (29.3 ± 5.8 years), runners (training: 38.5 ± 16.3 km/w)	No alcohol, caffeine, smoking, sunbathing, creams or body lotions, heavy meals, or exercise	Compression stocking	Controlled, not specified	No electronic equipment or persons	FLIR E-60 Accuracy: 2% Sensitivity: ≤0.05 °C	Anti-reflective panel	1 min
7	Priego-Quesada et al. [35]	19♂ (29.5 ± 9.8 years), runners (training: 229 ± 150 km/w)	No alcohol, caffeine, smoking, sunbathing, creams or body lotions, heavy meals, or exercise	NR	Controlled, not specified	No electronic equipment or persons, lights off	FLIR E-60 Sensitivity: ≤0.05 °C	Anti-reflective panel	1 min
8	Priego-Quesada et al. [36]	10♂ (25.0 ± 4.0 years), physically active cyclists	No alcohol, caffeine, smoking, sunbathing, creams or body lotion, heavy meals	NR	T°:19.5 ± 1.3 °C %RH:62.9 ± 3.2%	No electronic equipment or persons, lights and temperature controlled	FLIR E-60 Accuracy: 2% Sensitivity: ≤0.05 °C	Anti-reflective panel	10 min
9	Priego-Quesada et al. [37]	10♂ (40.0 ± 6.0 years), recreational triathletes (training: 7.0 ± 3.0 years of experience)	No alcohol, caffeine, smoking, sunbathing, creams or body lotion, heavy meals	No recovery protocols between training	T°:18 °C %RH:44–63%	No electronic equipment or persons, lights and temperature controlled	FLIR E-60 Accuracy: 2% Sensitivity: ≤0.05 °C	Anti-reflective panel	10 min (biped)
10	Hadžić, et al. [22]	1♂ (25 years), middle-distance runner (training: 12 years of experience)	NR	NR	T°: 23.5 ± 2 °C	NR	FLIR T425 Accuracy: 1%	NR	15 min
11	Fernandes et al. [38]	12♂ (22.4 ± 3.3 years), physically active (training: >3 times/w)	Controlled food intake. Avoid thermal stress, exercise.	NR	T°: 24.9 ± 0.6 °C %RH: 62.3 ± 5.7%	Artificial light	T420 Accuracy: 2% Sensitivity: ≤0.05 °C	NR	1 h
12	Rodriguez-Sanz et al. [39]	57♂ (42.8 ± 7.0 years), runners with functional equinus (training: 6 h/week)	No drugs or physical activity, heavy meals, or caffeine.	NR	T°: 24.1 ± 1.0 °C %RH: 45 ± 10%	Minimized airflow	FLIR SC3000	NR	NR
13	Priego-Quesada et al. [40]	9♂ (26 ± 10 years), recreational runners (training: (30.8 ± 28.9 km/w)	NR	NR	T°:23.2 ± 0.9 °C %HR: 27.9 ± 5.1%	NR	FLIR E60bx Accuracy: 2% Sensitivity: ≤0.05 °C	NR	5–10 min
14	Merla et al. [41]	15♂ (25.2 ± 3.1 years), well-trained runners	Shaved, no creams or lotions, no smoking, alcohol, or caffeine intake	NR	T°: 23–24 °C %HR: 50 ± 5%.	NR	FLIR SC3000	NR	20 min
15	Luo et al. [42]	19♂ (26.2 ± 1.2 years) runners	No exercise 24 h before	NR	T°: 26 °C %HR: 60–70%	NR	DaLi DL700	NR	NR
16	Rynkiewicz et al. [43]	14♂ and 5♀ (15.4 ± 1 years), kayakers (training experience: 4 years)	NR	NR	T°: 22 ± 0.5 °C %HR 40 ± 2%.	NR	ThermaCAM SC 640	NR	15 min
17	Robles-Dorado [44]	28♂ and 4 ♀ (38.8 ± 6.4 years), runners (training experience: 8.5 years)	NR	Same socks	NR	NR	PCE-TC 3 Accuracy: 2% Sensitivity: ≤2 °C	NR	NR
18	Sanz-López et al. [45]	20♂ (22.8 ± 4.2 years), active (training experience: 3 d/week)	No exercise	NR	T°: 21 °C %RH: 60%	NR	FLIR E60	NR	10 min
19	Priego-Quesada et al. [46]	16♂ (20 ± 10 years), club level cyclists	No alcohol, caffeine, smoking, sunbathing or UV rays, creams or body lotion, heavy meals, no exhaustive exercise	NR	T°: 23.4–24 °C %RH: 40.7–50.8%	No electronic equipment or persons, lights and temperature controlled	FLIR E60bx Accuracy: 2% Sensitivity: ≤0.05 °C	Anti-reflective panel	10 min
20	Ludwig et al. [47]	7♂ (20 ± 10 years), elite cyclists (training level: 66.89 VO_2_max = mL/kg/min)	No strenuous exercise	NR	T°: 22–23 °C %RH: 50%	No direct airflow, controlled light.	AVIO TVS700	NR	10 min
21	Priego-Quesada et al. [48]	11♂ (31 ± 7.4 years) cyclist (training: 264.5 km/week) and 11♂ (27.2 ± 6.6 years) non- cyclists	No alcohol, caffeine, smoking, sunbathing or UV rays, creams or body lotion, heavy meals, no exhaustive exercise	NR	T°: 23.5 ± 1.2 °C %RH: 49.9 ± 3.9%	No direct airflow, no electronic equipment near, controlled light.	FLIR T420 Sensitivity: <0.045 °C	Anti-reflective panel	10 min
22	Priego-Quesada et al. [49]	17♂ and 5♀ (34 ± 5 years) endurance runners (training: 36.6 ± 12.9 km∕week)	No alcohol, caffeine, smoking, sunbathing or UV rays, creams or body lotion, heavy meals, no exhaustive exercise	NR	T°: 22.9 ± 1.3 °C %RH: 44.4 ± 12.1%	No direct airflow, no electronic equipment near, controlled light.	FLIR E60 Accuracy: 2% Sensitivity: ≤0.05 °C	Anti-reflective panel	10 min
23	Jiménez-Pérez et al. [50]	15♂ (28 ± 7 years) and 15♀ (35 ± 7 years) recreational runners (training: 20 km/week)	No alcohol, caffeine, smoking, sunbathing or UV rays, creams or body lotion, heavy meals, no exhaustive exercise	NR	T°: 20.9 ± 1 °C %RH: 39.4 ± 6.4%	No direct airflow, no electronic equipment near, controlled light.	FLIR E60bx	Anti-reflective panel	10 min
24	Mendonca-Barboza et al. [51]	10♂ (23.5 ± 5.1 years) middle- distance elite runners (training: 2–3 h/d, 5 d/week)	No alcohol, smoking, drugs, or exercise was allowed	NR	T°: 22–24 °C %RH: 50%	NR	FLIR T360	NR	15 min
25	Duygu et al. [52]	18♂ sailing athletes	NR	NR	T°: 21 °C	NR	FLIR E5	NR	10–20 min
26	Pérez-Guarner et al. [53]	11♂ and 6♀ (41 ± 6 years) runners (training 5.9 ± 1.9 sessions/week)	No alcohol, caffeine, smoking, sunbathing or UV rays, creams or body lotion, heavy meals, no exhaustive exercise	NR	T°: 23.2 ± 0.1 °C %RH: 20 ± 1%	No direct airflow, no electronic equipment near, controlled light.	FLIR E60 Accuracy: 2% Sensitivity: ≤0.05 °C	Anti-reflective panel	10 min
27	Drzazga et al. [54]	6♂ cross-country skiers (23 ± 2.7 years) and 4♂ (21.5 ± 2.1 years) elite swimmers	NR	NR	T°: 19 ± 0.5 C %RH: 56 ± 3%	NR	FLIR E60 Sensitivity: 0.05 K	NR	1–2 min
28	Trecroci et al. [55]	10♂ (21.4 ± 2.6 years) elite cyclists	No strenuous exercise, no medication, drugs, cosmetic products, or caffeine intake.	NR	T°: 22–23 °C %RH: 50 ± 5%	Constant natural and fluorescent lighting and no direct ventilation	AVIO TVS-700	Constant temperature panel	10 min
29	Novotny et al. [56]	13 active students	No exercise	NR	T°: 27.9–28.1 °C %RH: 52.3–52.8%	NR	FLUKE TiR	NR	15 min
30	Novotny et al. [57]	25♂ (20.6 ± 1.61) active students	NR	NR	T°: 27.9–28.1 °C %RH: 52.3–52.8%	NR	FLUKE TiR Sensitivity: 0.1°	NR	15 min
31	Priego-Quesada et al. [58]	16♂ endurance runners	No alcohol, caffeine, smoking, sunbathing or UV rays, creams or body lotion, heavy meals, no exhaustive exercise	NR	T°: 27.9–28.1 °C %RH: 52.3–52.8%	No direct airflow, no electronic equipment near, controlled light.	FLIR E60bx Accuracy: 2% Sensitivity: ≤0.05 °C	Anti-reflective panel	10 min
32	Requena-Bueno et al. [59]	20♂ and 10♀ (34 ± 10 years) runners (training 34.6 ± 19.5 km/week)	No alcohol, caffeine, smoking, sunbathing or UV rays, creams or body lotion, heavy meals, no exhaustive exercise	NR	T°: 21.4 ± 2.0 °C %RH: 40.6 ± 10.1%	No direct airflow, no electronic equipment near, controlled light.	FLIR E60bx Accuracy: 2% Sensitivity: ≤0.05 °C	Anti-reflective panel	10 min
33	Bertucci et al. [60]	2♂ (16 years) competitive cyclists	NR	NR	NR	NR	Cedip Titanium HD560M	NR	NR
34	Ferreira-Oliveira et al. [61]	16♂ (22.5 ± 2.1) young active men	No diuretics, smoking, alcohol, or drugs consumption, dermatological treatments, or lotions	No skin burns, kidney problems, symptoms of pain, osteomioarticular injury	T°: 19.7 ± 1.5 °C %RH: 56.9 ± 5%	NR	Fluke ITR-25	NR	10 min
35	Andrade-Fernandes et al. [62]	12♂ (22.4 ± 3.3 years) active males (training: +3 times/week)	No diuretics, smoking, alcohol, or drugs consumption, dermatological treatments, or lotions	No skin burns, kidney problems, symptoms of pain, osteomioarticular injury	T°: 24.9 ± 0.6 °C %RH: 62.3 ± 5.7%	NR	FLIR T420	NR	30 min
36	Akimov & Son’kin [63]	20♂ (23.3 ± 4.8 years), athletes	NR	NR	T°: 21–22 °C %RH: 40%	NR	NEC TH9100	NR	10 min
37	Cholewka et al. [64]	12♂ (23.3 ± 4.8 years), cyclist	NR	NR	T°: 21.8 ± 1 °C	NR	FLIR E60 Sensitivity: 0.05 K	NR	30–40 min
38	Tanda [65]	6♂ and 1♀ (18–59 years), middle-long distance runners (training: 3–5 sessions/w)	NR	NR	T°: 22 °C %RH: 4%	Heating/cooling air conditioning system	FLIR T335	NR	10 min
39	Crenna & Tanda [66]	6♂	NR	NR	T°: 23–27 °C %RH: 60–70%	NR	FLIR T335	NR	10 min
40	Rojas-Valverde et al. [67]	9♂ and 7♀ (36 ± 7 years), distance runners (training: 10 ± 7 years of experience)	No drugs, alcohol, caffeine, smoking, exercise, lotions, or creams.	No neuromuscular injury or pathological or metabolic disease	T°: 23.0 ± 0.5 °C %RH: 58.0 ± 6%	Heating/cooling air conditioning system and radiation avoided	FLIR T440 Accuracy: 2% Sensitivity: 0.04 °C	Anti-reflective panel	15 min
41	Fernández-Cuevas et al. [68]	15♂ (21.4 ± 2.6 years), physically active college students	No drugs, alcohol, smoking, exercise.	NR	T°: 20.6 ± 0.7 °C %RH: 44.0 ± 3.2%	NR	FLIR T335	NR	15 min
42	Racinais et al. [69]	47♂ and 36 ♀(only 49 in Tsk), elite athletes	NR	NR	T°: 29.3 ± 0.5–32.7 ± 0.2 °C %RH: 46.3 ± 1.0–80.6 ± 1.1%	Skin was towel dried	FLIR T600	NR	NR
43	Machado et al. [70]	12♂ (25 ± 8 years), active males (training: 43.4 ± 44.2 km/week)	NR	NR	T°: 3.1 ± 0.9 °C %RH: 28.1 ± 5.1%	No electronic equipment or persons, lights and temperature controlled	FLIR E60bx, C2 and Flir-One-Pro LT	Anti-reflective panel	10 min
44	Binek et al. [71]	10♂ and 6 ♀(22 ± 3.23 and 23.7 ± 3.15 years), cross-country skiers	NR	NR	T°: 20 ± 1 %RH: 56 ± 3%	NR	FlirE60	NR	NR
45	Jones et al. [72]	Two♂ (16 and 18 years) middle-distance runners	NR	NR	T°: 28.2 ± 2.8 °C %RH: 43.0 ± 11.4%	NR	Flir T600		NR

**Table 3 life-11-01286-t003:** Articles’ methodological (camera settings and setup) considerations following the Thermographic Imaging in Sports and Exercise Medicine [28] checklist.

#	Reference	Camera Preparation	Image Recording	Camera Position	Emissivity	Assessment Time	Body Position	Method of Drying the Skin	Image Evaluation
1	Tumilty et al. [29]	NR	NR	Perpendicular	95%	Same time (as close as waking)	Biped	NR	AVG Tsk
2	Gil-Calvo et al. [30]	Turned up 10 min before	Distance: 1 m	Perpendicular	0.98	NR	Prone	NR	AVG Tsk
3	Gutiérrez-Vargas et al. [31]	NR	Distance: 3 m Height: 45 cm	Perpendicular	0.98	9 a.m.	Biped	NR	AVG Tsk
4	Fournet et al. [32]	NR	Distance: 1.9 m	NR	0.98	NR	Biped	NR	AVG Tsk
5	Priego-Quesada et al. [33]	Calibration	Distance: 1 m	Perpendicular	0.98	NR	Biped	NR	AVG Tsk
6	Priego-Quesada et al. [34]	Calibration	Distance: 1 m	Perpendicular	0.98	Same time (not specified)	Biped	NR	AVG Tsk
7	Priego-Quesada et al. [35]	Calibration Turned up 10 min before	Distance: 1 m	Perpendicular	0.98	Same time (not specified)	Biped	NR	AVG Tsk
8	Priego-Quesada et al. [36]	Calibration Turned up 10 min before	Distance: 1 m	Perpendicular	0.98	NR	Biped	NR	AVG Tsk
9	Priego-Quesada et al. [37]	Calibration Turned up 10 min before	Distance: 1 m	Perpendicular	0.98	7:45 a.m.	Biped	NR	AVG and MAX Tsk
10	Hadžić et al. [22]	Calibration	Distance: 1 m	NR	0.98	NR	Cycling	NR	AVG Tsk
11	Fernandes et al. [38]	NR	Distance: 1 m	NR	0.98	2 p.m.	Biped	NR	Median Tsk
12	Rodriguez-Sanz et al. [39]	NR	NR	Perpendicular	NR	NR	Biped	NR	AVG, MIN, MAX Tsk
13	Priego-Quesada et al. [40]	Calibration	Distance: 1.5 m	Perpendicular	0.98	NR	Biped	NR	AVG Tsk
14	Merla et al. [41]	NR	Distance: 4 m	NR	NR	Same time (late morning)	NR	NR	AVG Tsk
15	Luo et al. [42]	NR	NR	NR	NR	NR	Biped	NR	AVG, MIN, MAX Tsk
16	Rynkiewicz et al. [43]	NR	Distance: 6 m	Perpendicular	NR	NR	Biped	NR	AVG Tsk
17	Robles-Dorado [44]	NR	NR	Perpendicular	NR	NR	Biped/supine	NR	AVG Tsk
18	Sanz-López et al. [45]	NR	Distance: 2.5 m	NR	NR	NR	Biped	NR	AVG Tsk
19	Priego-Quesada et al. [46]	NR	Distance: 1 m	Perpendicular	NR	Same Time (not specified)	Biped	Sweat removed (not specified)	AVG Tsk
20	Ludwig et al. [47]	Fixed in a tripod	NR	Perpendicular	0.98	NR	Biped	NR	AVG, MAX Tsk
21	Priego-Quesada et al. [48]	NR	Distance: 1 m	Perpendicular	NR	NR	Biped	NR	AVG Tsk
22	Priego-Quesada et al. [49]	Calibration	Distance: 1 m	Perpendicular	NR	NR	Biped	NR	AVG Tsk
23	Jiménez-Pérez et al. [50]	Calibration	Distance: 1 m	Perpendicular	NR	NR	Supine	NR	AVG Tsk
24	Mendonca-Barboza et al. [51]	Fixed in a tripod	Distance: 3.5	Perpendicular	NR	Same time (afternoon)	Biped	NR	AVG Tsk
25	Duygu et al. [52]	NR	Distance: 1 m	NR	NR	NR	Biped	NR	MAX Tsk
26	Pérez-Guarner et al. [53]	Calibration	Distance: 1.5 m	Perpendicular	NR	Same time (afternoon)	Biped	NR	AVG, MAX Tsk
27	Drzazga et al. [54]	NR	NR	NR	NR	NR	NR	NR	AVG Tsk
28	Trecroci et al. [55]	NR	NR	NR	0.98	Same time (morming)	Biped	NR	MAX Tsk
29	Novotny et al. [56]	NR	NR	NR	0.98	NR	Biped	NR	AVG Tsk
30	Novotny et al. [57]	NR	NR	NR	0.98	NR	Biped	NR	AVG Tsk
31	Priego-Quesada et al. [58]	NR	Distance: 1 m	Perpendicular	0.98	NR	Biped	NR	AVG Tsk
32	Requena-Bueno et al. [59]	Stabilisation	Distance: 1 m	Perpendicular	0.98	NR	Supine	NR	AVG Tsk
33	Bertucci et al. [60]	NR	NR	NR	NR	NR	NR	NR	NR
34	Ferreira-Oliveira et al. [61]	NR	NR	NR	NR	NR	Biped	NR	AVG Tsk
35	Andrade-Fernandes et al. [62]	NR	Distance 3 m	NR	0.98	Same time (afternoon)	Running/Biped	NR	AVG Tsk
36	Akimov & Son’kin [63]	NR	NR	NR	NR	NR	Cycling	NR	AVG Tsk
37	Cholewka et al. [64]	NR	NR	NR	NR	NR	Cycling	NR	Curve Tsk
38	Tanda [65]	Calibration	NR	NR	0.98	Same time (late morning)	Running	NR	AVG Tsk
39	Crenna & Tanda [66]	Calibration	Distance: 3 m	NR	NR	Same time (late morning)	Running	NR	AVG Tsk
40	Rojas-Valverde et al. [67]	Camera turned on 30 min before each test	Distance: 3 m Height 60 cm, 5° angle	Perpendicular	0.98	Same time (7:00–7:30 a.m.)	Biped	Clean with water and then dried	AVG Tsk
41	Fernández-Cuevas et al. [68]	NR	NR	NR	NR	8:30–11:30 a.m.	Biped	NR	AVG Tsk
42	Racinais et al. 2021 [69]	Accuracy checked using 20 × 20 cm reference plates	Distance: 4 m	NR	0.98	NR	Biped	Towel dried	AVG Tsk
43	Machado et al. [70]	Calibration	Distance: 3 m	NR	0.98	NR	Seated, biped	NR	MAX and AVG Tsk
44	Binek et al. [71]	Calibration	NR	NR	0.98	NR	Biped	NR	AVG Tsk
45	Jones et al. [72]	NR	1.5 m	NR	NR	NR	Biped	NR	AVG Tsk

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
