# Peer review of "Short-Term Skin Temperature Responses to Endurance Exercise: A Systematic Review of Methods and Future Challenges in the Use of Infrared Thermography"

_life, 2021, doi:10.3390/life11121286_

Round 1

Reviewer 1 Report

The manuscript presents a systematic review in short-term monitoring responses to endurance exercise using infrared thermography and analysing the method of application and future challenges.
Results showed that ore analyses should be made considering different sports, exercise stimuli and intensities, especially using follow-up designs. Study derived data could clarify the underlying physiological processes involved in the regulation of skin temperature and its interactions with other tissues during exercise-related responses such as inflamma34 tion, damage, or pain.

I find the topic interesting and being worth of investigation and the document is well strucutred, organized, fluidly written, the methodology well described and reproducible, the results are clearly presented and support the conclusions.
Although I propose the following suggestions / considerations:
- Abstract requires structuring such as: problem, motivation, aim, methodology, main results, further impact of those results.
- Keywords should be in alphabetical order.

Author Response

Dear Editor and reviewers:

We have carefully considered all reviewers' recommendations for the paper (Life-1439346) entitled " Short-term skin temperature responses to endurance exercise: a systematic review of methods and future challenges in the use of infrared thermography”. Please find enclosed our detailed answers to reviewers' queries. The authors declare that the manuscript is original and has not been considered for publication elsewhere. Additionally, the authors had approved the paper for release and agreed with its content.

Please find all corrections in red inside the manuscript.

Reviewer: 1

R1.1. The manuscript presents a systematic review in short-term monitoring responses to endurance exercise using infrared thermography and analysing the method of application and future challenges.

R/ The authors wish to thank the reviewers for their time and dedication in reviewing this manuscript. We have considered all of your recommendations and made any necessary adjustments and corrections. We trust that these modifications have had a positive impact on the final quality of the manuscript.

R1.2.Results showed that ore analyses should be made considering different sports, exercise stimuli and intensities, especially using follow-up designs. Study derived data could clarify the underlying physiological processes involved in the regulation of skin temperature and its interactions with other tissues during exercise-related responses such as inflammation, damage, or pain.

R/We want to thank you for your time and dedication.

R1.3.I find the topic interesting and being worth of investigation and the document is well strucutred, organized, fluidly written, the methodology well described and reproducible, the results are clearly presented and support the conclusions.

R/ R/We want to thank you for your time and dedication.

Although I propose the following suggestions / considerations:
R1.4.- Abstract requires structuring such as: problem, motivation, aim, methodology, main results, further impact of those results.

R/ We have corrected the abstract following this recommendation and other reviewer´s insights. We followed the journal guidelines to select the sub-headings.

R1.5.- Keywords should be in alphabetical order.

R/The keywords were changes according to this recommendation and other reviewer´s insights.

Reviewer: 2

R2.1 The paper presents a systematic review of the application of infrared thermography in endurance sports. The paper is well written and discussed. I do not acknowledge a review that covers this method with the specific application for endurance exercise. The theme is recent with the evolution of thermal cameras, and several studies have been published in the last years. I found interesting in the discussion the concern with the standardization of methods and correct information for the reproducibility of the experiments. The article can serve as a guide for studies of physical exercise to define its experimental procedures.

R/ The authors wish to thank the reviewers for their time and dedication in reviewing this manuscript. We have considered all of your recommendations and made any necessary adjustments and corrections. We trust that these modifications have had a positive impact on the final quality of the manuscript.

R2.2.Some of the works among those analyzed were prepared by the same research group from some institutions. Would it be possible to identify and comment on them since these works might use very similar methodologies?

R/ This recommendation was addressed in the new version of the text.

Below are some formatting suggestions:
R2.3.- Table 1: Include year of publication in the second column.

R/As recommended we have added the year of publication in the second column

R2.4.- Table 2 and 3: Could include article number (1st column of table 1) to help locate between tables.

R/As recommended we have added the article number in first column

Reviewer: 3

R3.1. The present manuscript is a systematic review on infrared methods used in endurance events. It is an interesting topic with growing importance in exercise and thermal physiology. However, the ms has several important limitations or missing parts which must be addressed.

R/ The authors wish to thank the reviewers for their time and dedication in reviewing this manuscript. We have considered all of your recommendations and made any necessary adjustments and corrections. We trust that these modifications have had a positive impact on the final quality of the manuscript.

ABSTRACT:

R3.2 The following sentences: “The common thermo physiological response of the Tsk is to decrease during endurance exercise and increase during recovery. An inverse relationship between core and Tsk was found during exercise with several con textual and situational factors influencing Tsk changes. »

R/We have corrected the abstract following this recommendation and other reviewer´s insights.

R3.3 Are results extracted from the studies considered and not really related to the methodology aspects of the ms. This should not appear here.

R/We have corrected the abstract following this recommendation and other reviewer´s insights.

KEYWORDS:

R3.4 Why Load monitoring? It would be more appropriate to introduce a keyword like “heat stress” or heat stroke”

R/The keywords were changes according to this recommendation and other reviewer´s insights.

INTRODUCTION:

Two crucial points are missing in the Introduction section:

R3.5.Global warming and the increasing risk of mass/elite events being organised in hot/humid environments. This should be developed.

R/We agree with the reviewer the section was changed.

R3.4.No reference at all to the major problem here: exertional heat stroke and its deleterious consequences for health. I don’t think the structural modification of the tendon and the muscle are a concern here. References 10, 11, 12, 13 are not interesting. Please develop the heat illnesses-related aspects and introduce corresponding recent references (from Doug Casa team for instance).

R/We have rewrite the section based on the reviewer comments and recommendations.

R3.4.As a result, the paragraph between lines 64-70 is to be deleted or significantly reduced.

R/We have deleted the paragraph based on the reviewer recommendation.

METHODS:

R3.5. I am not fully convinced about the keyword choice decided by the authors. Indeed, a significant and recent article from Racinais et al (Br J Sports Med doi:10.1136/bjsports-2020-103613) is missing from your selection whereas it is exactly in the scope of your review. I am afraid this is not the only one. Could you please rethink your keywords choice and rerun your research and analysis?

R/We have re-run the systematic search and we made a new analysis based on the findings. We missed some studies due to their publication date. These four studies were now included in the analysis. Consequently the figure 1. changed and it was improved. Also results were rewritten and recalculated. Discussion was adjusted based on these new results.

R3.6. Line 130: what do you mean by acclimation here. In thermophysiology acclimation has a specific meaning. Please clarify.

R/We agree with the reviewer the term was changed to avoid confusion, instead thermal adaptation was used based on Machado et al. [70]

RESULTS:

R3.7. This whole section needs to be better presented as it is not reader friendly. I would suggest that below each subsection (3.1,…) you identify the single parameter (from the TISEM consensus statement, for instance ROI,…) you analysed in your x selected articles and introduce the results (absolute and relative).

R/The results were modified according to the reviewer consideration.

R3.8 Line 147-148: What do you mean by transversal design. My understand of a transversal design is that there is no pre-post measurement like mentioned in line 148. Can you please clarify. From info given lines 149-154, I counted 33 studies instead of 38. Where are the 5 missing studies?

R/We appreciate the reviewer for the opportunity to clarify, there is now 45 articles included in the systematic review. Also, we change the term transversal design for cross-sectional.

R3.9 Line 158: what is measurement moment? What is “some considerations”? What about measuring front and back of the subjects? Could you include this in the parameter measured

R/We appreciate the reviewer for the opportunity to clarify, we have change the term measurement moment for time-point. The information was added as requested throughout the manuscript.

R3.10 Lines 168-173: these are results from the studies you analysed. Not sure this should appear here, since you are discussing methodology of the papers and not their results.

R/The outcomes indicators is part of the TISEM guidelines, so we kept it as a basic description.

R3.11 Line 173: isn’t it risky to make a statement (sex) based on a single article.

R/we have corrected this statement.

R3.12 Line 177: what’s the reason for moving from 41 studies to 35. How did you select these 35 studies? Please explain.

R/This issue was a confusion, there is no so statement.

R3.13 Line 181 and beyond: as explained before please ask subchapter titles here (parameter analysed) to improve the readability.

R/Results section was clarified and rewritten

R3.14 Line 207: define calibration. Is it using a reference plate connected to a thermistor?

R/The calibration was clarified.

DISCUSSION:

R3.15 Again, what does transversal through pre post mean?

R/This have been changed, according to the reviewer recommendation

R3.16 This chapter should be completely rewritten. Indeed, we still find in the actual Discussion section some “Results” aspects (for instance lines 227-232).

R3.17 The” subchapter 4.1 should be rewritten discussing the obtained results with the TISEM Consensus Statement.

R3.18 The present ms is systematic review of Methods! Therefore, what is the purpose of chapter 4.2? It is not a systematic review of Results. The whole chapter should be removed.

R/The discussion was rewritten as recommended. The section 4.2 was changed.

R3.19 Line 283-284: the mechanical damage resulting from the excitation contraction coupling failure is NOT the point here. It is heat illnesses and exertional heat stroke. Developing some future affordable and reliable skin temp sensors to better predict these (sometime) lethal condition is a perspective that must be discussed.

R/The discussion was rewritten as recommended. The section 4.2 was changed.

R3.20 I am surprised that the following article is not even cited in the Discussion: Aylwin PE, Racinais S, Bermon S, Lloyd A, Hodder S, Havenith G. The use of infrared thermography for the dynamic measurement of skin temperature of moving athletes during competition; methodological issues. Physiol Meas. 2021 Aug 27;42(8). doi: 10.1088/1361-6579/ac1872. PMID: 34320480. This would be a great opportunity to introduce in your rewritten discussion/limitation/perspective the relevance of high frequency dynamic thermal camera.

R/The discussion was rewritten as recommended. The section 4.2 was changed.

CONCLUSION

R3.21 Considering, the above recommendation, the conclusion should be rewritten.

R/The conclusion section was rewritten as recommended.

Reviewer 2 Report

The paper presents a systematic review of the application of infrared thermography in endurance sports. The paper is well written and discussed. I do not acknowledge a review that covers this method with the specific application for endurance exercise. The theme is recent with the evolution of thermal cameras, and several studies have been published in the last years. I found interesting in the discussion the concern with the standardization of methods and correct information for the reproducibility of the experiments. The article can serve as a guide for studies of physical exercise to define its experimental procedures.

Some of the works among those analyzed were prepared by the same research group from some institutions. Would it be possible to identify and comment on them since these works might use very similar methodologies?

Below are some formatting suggestions:
- Table 1: Include year of publication in the second column.
- Table 2 and 3: Could include article number (1st column of table 1) to help locate between tables.

Author Response

(The authors gave the same response as above.)

Reviewer 3 Report

The present manuscript is a systematic review on infrared methods used in endurance events. It is an interesting topic with growing importance in exercise and thermal physiology. However, the ms has several important limitations or missing parts which must be addressed.

ABSTRACT:

The following sentences: “The common thermo-27 physiological response of the Tsk is to decrease during endurance exercise and increase during recovery. An inverse relationship between core and Tsk was found during exercise with several con-29 textual and situational factors influencing Tsk changes. »

Are results extracted from the studies considered and not really related to the methodology aspects of the ms. This should not appear here.

KEYWORDS:

Why Load monitoring? It would be more appropriate to introduce a keyword like “heat stress” or heat stroke”

INTRODUCTION:

Two crucial points are missing in the Introduction section:

  1. Global warming and the increasing risk of mass/elite events being organised in hot/humid environments. This should be developed.
  2. No reference at all to the major problem here: exertional heat stroke and its deleterious consequences for health. I don’t think the structural modification of the tendon and the muscle are a concern here. References 10, 11, 12, 13 are not interesting. Please develop the heat illnesses-related aspects and introduce corresponding recent references (from Doug Casa team for instance).
  3. As a result, the paragraph between lines 64-70 is to be deleted or significantly reduced.

METHODS:

I am not fully convinced about the keyword choice decided by the authors. Indeed, a significant and recent article from Racinais et al (Br J Sports Med doi:10.1136/bjsports-2020-103613) is missing from your selection whereas it is exactly in the scope of your review. I am afraid this is not the only one. Could you please rethink your keywords choice and rerun your research and analysis?

Line 130: what do you mean by acclimation here. In thermophysiology acclimation has a specific meaning. Please clarify.

RESULTS:

This whole section needs to be better presented as it is not reader friendly. I would suggest that below each subsection (3.1,…) you identify the single parameter (from the TISEM consensus statement, for instance ROI,…) you analysed in your x selected articles and introduce the results (absolute and relative).

Line 147-148: What do you mean by transversal design. My understand of a transversal design is that there is no pre-post measurement like mentioned in line 148. Can you please clarify. From info given lines 149-154, I counted 33 studies instead of 38. Where are the 5 missing studies?

Line 158: what is measurement moment? What is “some considerations”? What about measuring front and back of the subjects? Could you include this in the parameter measured

Lines 168-173: these are results from the studies you analysed. Not sure this should appear here, since you are discussing methodology of the papers and not their results.

Line 173: isn’t it risky to make a statement (sex) based on a single article.

Line 177: what’s the reason for moving from 41 studies to 35. How did you select these 35 studies? Please explain.

Line 181 and beyond: as explained before please ask subchapter titles here (parameter analysed) to improve the readability.

Line 207: define calibration. Is it using a reference plate connected to a thermistor?

DISCUSSION:

Again, what does transversal through pre post mean?

This chapter should be completely rewritten. Indeed, we still find in the actual Discussion section some “Results” aspects (for instance lines 227-232).

The” subchapter 4.1 should be rewritten discussing the obtained results with the TISEM Consensus Statement.

The present ms is systematic review of Methods! Therefore, what is the purpose of chapter 4.2? It is not a systematic review of Results. The whole chapter should be removed.

Line 283-284: the mechanical damage resulting from the excitation contraction coupling failure is NOT the point here. It is heat illnesses and exertional heat stroke. Developing some future affordable and reliable skin temp sensors to better predict these (sometime) lethal condition is a perspective that must be discussed.

I am surprised that the following article is not even cited in the Discussion: Aylwin PE, Racinais S, Bermon S, Lloyd A, Hodder S, Havenith G. The use of infrared thermography for the dynamic measurement of skin temperature of moving athletes during competition; methodological issues. Physiol Meas. 2021 Aug 27;42(8). doi: 10.1088/1361-6579/ac1872. PMID: 34320480. This would be a great opportunity to introduce in your rewritten discussion/limitation/perspective the relevance of high frequency dynamic thermal camera.

CONCLUSION

Considering, the above recommendation, the conclusion should be rewritten.

Author Response

(The authors gave the same response as above.)

Reviewer 4 Report

The manuscript presents a systematic review on the application of infrared thermography imaging in detecting short-term skin temperature (Tsk) responses to endurance exercise and to identifying the methodological considerations and knowledge gaps, allowing the proposing of future
research directions.

Results show that more analyses should be made considering different sports, exercise stimuli and intensities, especially using follow-up designs.

At the methodology using just the term exercise, it may be very general and not specific of any kind of sport activity.

The keywords should be in alphabetical order.

I would advise authors to enlarge the surveying key terms to add more knowledge, because after analyzing the outcomes of their review a little or almost nothing is added to the existing knowledge.

Aspects such as imaging protocols, equipment characterization, participants preparation and temperature assessment techniques may have greater importance for outlining future directions.

Round 2

Reviewer 3 Report

I thank the authors for taking into account my suggestions and recommendations. The ms reads much better now.

However, there are two remaining points which need to be amended or modified, before being, on my humble opinion, considered suitable for publication.

1°) ABSTRACT:

The conclusion last sentence could be much improved replacing the actual one by something like: " Study-derived data could clarify the underlying thermophysiological processes and assess whether Tsk could be used a reliable proxy to describe live thermal regulation in endurance athletes and reduce their risk of exertional heat illness/stroke."

2°) INTRODUCTION :

I don't see the relationship between the sentence l 71-72 and the following one referring to eccentric action and mechanical injuries. I would suggest to delete most of this last sentence only emphasising on heat issues.

Reviewer 4 Report

I am happy with the author's improvements and answers, please accept the manuscript for publication.